# Intranasal Exposure to Rift Valley Fever Virus Live-Attenuated Strains Leads to High Mortality Rate in Immunocompetent Mice

**DOI:** 10.3390/v14112470

**Published:** 2022-11-08

**Authors:** Sandra Lacote, Carole Tamietti, Mehdi Chabert, Marie-Pierre Confort, Laurine Conquet, Coralie Pulido, Noémie Aurine, Camille Baquerre, Adrien Thiesson, Bertrand Pain, Marcelo De Las Heras, Marie Flamand, Xavier Montagutelli, Philippe Marianneau, Maxime Ratinier, Frédérick Arnaud

**Affiliations:** 1ANSES, Lyon Laboratory, Virology Unit, 69007 Lyon, France; 2Institut Pasteur, Structural Virology, Université Paris Cité, 75012 Paris, France; 3IVPC UMR754, INRAE, Univ Lyon, Université Claude Bernard Lyon 1, EPHE, PSL Research University, 69007 Lyon, France; 4Mouse Genetics Laboratory, Institut Pasteur, Université Paris Cité, 75015 Paris, France; 5ANSES-Laboratoire de Lyon, Plateforme d’Expérimentation Animale, 69007 Lyon, France; 6INSERM, INRAE, Univ Lyon, Stem Cell and Brain Research Institute, U1208, USC1361, Université Lyon 1, 69500 Bron, France; 7Departamento de Patología Animal, Instituto Agroalimentario de Aragón-IA2 (Universidad de Zaragoza-CITA), Facultad de Veterinaria, 50013 Zaragoza, Spain

**Keywords:** Rift Valley fever virus, attenuated vaccine strains, intranasal exposure, pathogenicity

## Abstract

Rift Valley fever virus (RVFV) is a pathogenic arthropod-borne virus that can cause serious illness in both ruminants and humans. The virus can be transmitted by an arthropod bite or contact with contaminated fluids or tissues. Two live-attenuated veterinary vaccines—the Smithburn (SB) and Clone 13 (Cl.13)—are currently used during epizootic events in Africa. However, their residual pathogenicity (i.e., SB) or potential of reversion (i.e., Cl.13) causes important adverse effects, strongly limiting their use in the field. In this study, we infected immunocompetent mice with SB or Cl.13 by a subcutaneous or an intranasal inoculation. Interestingly, we found that, unlike the subcutaneous infection, the intranasal inoculation led to a high mortality rate. In addition, we detected high titers and viral N antigen levels in the brain of both the SB- and Cl.13-infected mice. Overall, we unveil a clear correlation between the pathogenicity and the route of administration of both SB and Cl.13, with the intranasal inoculation leading to a stronger neurovirulence and higher mortality rate than the subcutaneous infection.

## 1. Introduction

Rift Valley fever virus (RVFV) is a zoonotic and hemorrhagic fever virus that infects both ruminants (mainly sheep and cattle) and humans. In the former, it causes high mortality rates in new-born animals and massive abortions in pregnant females (with, sometimes, a 100% prevalence) while, in the latter, it leads to milder symptoms characterized by a self-limiting, acute, febrile illness [1,2,3,4]. In 1 to 3% of cases, however, patients develop more severe diseases such as fulminant hepatitis associated with hemorrhage and, occasionally, retinitis or late-onset encephalitis, with a fatality rate in hospitalized patients that ranges from 10 to 20%, depending on the epidemics [5,6,7].

RVFV is an arthropod-borne virus (arbovirus) naturally transmitted by infected mosquitoes belonging to the *Aedes* and *Culex* genera, even though it can be found in an unusual broad spectrum of arthropod species, including blackflies (*Simulium* spp.), midges (*Culicoides* spp.), tropical bont ticks (*Amblyomma variegatum*), and several species of mosquitoes and sand flies (*Phlebotomus* spp.) [3,8,9]. The virus can also spread via aerosol or direct contact with the organs or fluids of infected animals. For these reasons, certain professions such as veterinarians, farmers, and slaughterhouse workers are at higher risk of infection [10,11].

The World Health Organization has classified RVFV in the top ten priority list of emerging pathogens likely to cause severe outbreaks in the future. Until recently, indeed, RVFV periodic epidemics mainly occurred in sub-Saharan Africa, however, in recent decades, due to climate changes, they extended to the Arabian Peninsula, causing major economic losses in animal husbandry and thousands of human deaths [12,13,14,15]. The wide range and abundance of its host species alongside the possibility of infection by aerosol exposure make RVFV a major threat for humans and livestock, as well as a potential agent of bioterrorism; vaccination is therefore key to prevent a viral spread. To date, however, no licensed RVFV vaccine or treatment has been developed for human use, while three veterinary vaccines are currently available on the market: one formalin-inactivated, derived from an ancient strain, and two live-attenuated vaccines—the Smithburn (SB) and the Clone 13 (Cl.13) strains—which are widely used in endemic countries during epizootics [16,17]. The SB strain was the first vaccine to be developed against RVFV, in 1949, via the serial intracerebral passages of the pathogenic Entebbe strain in mice [11]. It is mainly used in eastern and southern Africa, and it is quite effective as it is able to induce a long-term protective immunity with a single dose [17]. However, it is not very safe as it retains a residual pathogenicity that can provoke abortions and fetal malformations in pregnant females [18]. For these reasons, the SB vaccine strain is recommended only in non-pregnant animals in RVFV endemic countries before and after outbreaks [17]. The Cl.13 strain was isolated from a plaque clone of the RVFV 74HB59 strain recovered from a patient of the Central African Republic [19]. Since 2010, it is a registered veterinary vaccine, increasingly used in western Africa [16]. Interestingly, the Cl.13 strain displays a large deletion in the open reading frame of the viral non-structural protein NSs [19]. NSs is the main virulence factor of RVFV as it strongly antagonizes the type-I interferon synthesis and inhibits the general host transcription by perturbing the assembly of the transcription factor II human (TFIIH) complex [20]. The Cl.13 live-attenuated vaccine appears to be safer than SB, even if, at high doses, it causes abortions [21].

Mice represent good, convenient, and well-established animal models to study the routes of transmission, the mechanisms of spread to target organs, and virulence of RVFV [22]. Along this line, RVFV-infected mice display important similarities to severe human infections with, depending on the mouse genetic background, extensive and fatal liver damages or delayed-onset encephalitis [23]. Moreover, they share the same RVFV tissue tropism and target cells with humans. Indeed, RVFV infects human and murine dendritic cells and macrophages, which probably facilitates a virus dissemination to various organs, potentially through the blood vessels [24,25]. Subsequently, RVFV disseminates to several tissues, including the liver, where it infects the hepatocytes and stellate cells. Occasionally, the virus reaches the central nervous system, where it is detectable in both the neurons and astrocytes of infected mice [22]. Notably, RVFV can also target the olfactory neurons lining the nasal tract and, in mice, aerosol exposure to RVFV results in an earlier and more severe neuropathology compared to a subcutaneous (SC) inoculation [26,27,28,29]. Interestingly, a recent study demonstrated that, in cattle, the intranasal administration of a Kenyan pathogenic RVFV strain induces a higher and long-lasting viremia and stronger clinical signs compared to an intradermal or SC inoculation [30]. On the other hand, an attenuated RVFV lacking the NSs sequence has been shown to cause severe and lethal neurological diseases in immunocompetent mice, only when infected intranasally and not subcutaneously [31]. Moreover, one study has shown that the Cl.13 intranasally (IN)-infected mice developed lethal encephalitis within 11 days post-infection, although they did not compare the IN route with the SC inoculation [32]. Overall, little is known on the impact of the route of transmission on RVFVs pathogenicity, particularly on that of the SB and Cl.13 vaccine strains. Unveiling this process is very important, not only to better understand the mechanisms underlying the efficacy and safety of these vaccines, but also to assess whether an aerosol exposure to RVFV may impact its pathogenicity (such as in the case of laboratory exposure or a bioterrorist attack). In this study, we characterized the viral titers and tissue tropism of RVFV SB and Cl.13 following an IN or SC infection, thus providing an important insight into their virulence and dissemination in immunocompetent mice.

## 2. Materials and Methods

### 2.1. Cell Cultures

BSR were kindly provided by Prof Karl Conzelmann (Ludwig-Maximilians-University Munich, Gene Center, Munich/Germany). VeroE6 cells were purchased from ATCC. The BSR and VeroE6 cells were grown in Dulbecco’s modification of an Eagle medium (DMEM) (Gibco, Thermo Fisher Scientific, Villebon-sur-Yvette, France), supplemented with 10% heat-inactivated fetal bovine serum (FBS) (GE HEALTHCARE Europe GmbH, Freiburg, Germany), and 25 µg/mL of penicillin-streptomycin (Gibco). The HepaRG cells were purchased from Lonza (Colmar, France) and grown in William’s E medium (Gibco), supplemented with 10% heat-inactivated FBS, 1% L-glutamine (Gibco), 5 µg of insulin (Gibco), and 0.5 µM of hydrocortisone (Sigma-Aldrich, Merck, Saint-Quentin Fallavier, France). Human embryonic fibroblasts (HEF) were kindly provided by Prof Odile Boespflug-Tanguy (AP-HP, Robert Debre Hospital, Department of Neuropediatrics and Metabolic Diseases, National Reference Center for Leukodystrophies, Paris, France). The HEF were grown in a fibroblast medium containing DMEM/F-12 (Gibco, Thermo Fisher Scientific), 10% fetal calf serum (Gibco, Thermo Fisher Scientific), 1X penicillin-streptomycin (Gibco, Thermo Fisher Scientific), and 2 mM of L-glutamine (Gibco, Thermo Fisher Scientific). All the cell lines were cultured in a 37 °C, 5% CO_2_ humidified incubator.

### 2.2. Human Induced-Pluripotent Stem Cells

The HEF were reprogrammed into human-induced pluripotent stem cells (hiPSC) using a CytoTune™ Sendai Reprogramming Kit (Life Technologies, Thermo-Fisher, Illkirch-Graffenstaden, France), according to the manufacturer’s instructions. The HEF and hiPSCs were declared with the number DC-2020-3895 onto the CODECOH platform “https://appliweb.dgri.education.fr/appli_web/codecoh/IdentCodec.jsp (accessed on 18 May 2020)”. The cells were maintained in mTeSR1 (StemCell Technologies, Saint Egreve, France) on a matrigel coating (Corning, Thermo-Fisher, Illkirch-Graffenstaden, France), and dissociated with dispase (StemCell Technologies), according to the manufacturer’s instructions. Two independent isolates were used. These cells were subsequently differentiated into neurons, astrocytes, and oligodendrocytes, according to the protocol reported below. 

### 2.3. Induction into 2D Neural Differentiation

Once dissociated, the cells were seeded into a 12-well tissue culture plate (1.5 × 10^5^ cells/mL) and grown into the neural induction medium (NIM). NIM contains DMEM/F-12 (Thermo-Fisher, Illkirch-Graffenstaden, France) and it is complemented with 2 mM of L-glutamine (Thermo-Fisher), 1000 U/mL of penicillin-streptomycin (Thermo-Fisher), 1% MEM non-essential amino acids solution (Thermo-Fisher), 1 mM of 2-mercaptoethanol (Thermo-Fisher), and 1% N-2 supplement (Thermo-Fisher). The medium was changed every two days, and replaced, on day 7, with the neural stem medium, containing NIM supplemented with 20 ng/mL of human recombinant basic Fibroblast Growth Factor (hrFGF) (154 a.a., Peprotech, Neuilly-Sur-Seine, France), and 20 ng/mL of murine recombinant Epidermal Growth Factor (mrEGF) (Peprotech). The medium was changed every two days. At day 14, the medium was replaced with NIM supplemented with 0.5 uM of all-trans retinoic acid (ATRA) (Sigma-Aldrich Chimie, Saint Quentin Fallavier, France) for 4 days, and with a fresh media at day 16. At day 18, the NIM was complemented with 0.5 uM of ATRA, 2% B-27 supplement (Thermo-Fisher), and 100 ng/mL of human recombinant Sonic Hedge Hog (hrSHH) (StemCell Technologies, Saint Egreve, France). The cells were grown in this medium for 10 days, with a change every two days. At day 28, the medium was replaced with the NIM complemented with 2% B-27 supplement, 100 ng/mL of hrSHH, and 10 ng/mL of hrFGF. The cells were grown in this medium for 12 days, and the medium was changed every two days. At day 40, the medium was adjusted again for the cells’ maturation, and supplemented with 2% B-27 supplement, 100 ng/mL of hrSHH, 10 ng/mL of human recombinant Platelet derived growth factor-AA (Peprotech), and 40 ng/mL of 3,3′,5-Triiodo-L-thyronine sodium salt (Sigma-Aldrich). The medium was changed every two days until the cells were infected and analyzed.

### 2.4. Virus Stocks and Titrations

RVFV SB and Cl.13 attenuated vaccines were purchased from Onderstepoort Biological Products. All the experiments with infectious RVFV SB and Cl.13 strains were carried out in biosafety level (BSL)-3 laboratories. The RVFV SB and Cl.13 viral stocks titers were determined by standard plaque assays using serial 10-fold dilutions in VeroE6 cells. Briefly, the monolayers of VeroE6 cells were plated onto 12-wells plates and infected for 2 h at 37 °C. The inocula were then removed and the cells were washed twice with phosphate-buffered saline (PBS). After, one and a half milliliters of a semisolid overlay (2.5% Agarose in 2X MEM supplemented with 4% FBS) were added on the top of the cells, and the plate was incubated at 37 °C for 3 to 5 days. Finally, the overlay was removed, and the cells were washed with PBS before being fixed with 4% paraformaldehyde and stained either using an anti-N RVFV antibody (kindly provided by Dr Benjamin Brennan, MRC-University of Glasgow Centre for Virus Research-Glasgow) [33] or with 0.2% crystal violet, 3.7% formaldehyde, and 20% ethanol solution. The viral titers are expressed as FFU/mL or PFU/mL, respectively.

### 2.5. Virus Growth Curves

Prior to infection, the HepaRG and neural differentiated hIPSC were seeded into 12-well and 24-well tissue culture plates (Corning), respectively. The cells were then infected with the SB or Cl.13 strain at 0.01 (for HepaRG) and 0.1 (for hIPSC) multiplicity of infection (MOI) and incubated at 37 °C for 2 h. The cells were then washed three times with a fresh media and incubated with 1 mL of the appropriate fresh growth medium (herein, this time point is referred to as “T0” post-infection, p.i.). The culture supernatants (100 µL) were harvested at 24 h and 48 h p.i. and replaced with a fresh growth medium. The viral supernatants were titrated by limiting the dilution assays in BSR cells, as previously described [34]. Viral titers are expressed as 50% tissue culture infective doses (TCID50)/milliliter. All the experiments were performed independently in triplicate and repeated at least three times. Statistical analyses were conducted at 24 h and 48 h p.i. with viral titers equal to or above the threshold of detection (i.e., 1.5 log_10_ TCID_50_/mL), using the Kruskal–Wallis test and Graphpad Prism 8.4 (Graphpad Software Inc., La Jolla, CA, USA).

### 2.6. Mice Infection Assays

Six- to eight-week-old female BALB/cJRj mice were purchased from Janvier Labs (Le Genest St Isle, France). Groups of 5–6 mice were inoculated subcutaneously or intranasally with a 1 × 10^3^ plaque-forming unit (PFU) of the SB or Cl.13 viral strains. Subcutaneous inoculations were performed on unanesthetized mice in the ventral region under a 100 µL volume. Intranasal inoculations were performed on mice anaesthetized with an intraperitoneal injection of Ketamine (60 mg/kg) and Xylazine (2 mg/kg). Ten microliters of inoculum were instilled in each nare. Three animals were enrolled in PBS-inoculated control groups. The clinical signs and survival rates were recorded daily over 15 days p.i. Mice showing severe clinical signs (ruffled fur, hunched posture, loss of >12% of their initial body weight) were humanely euthanized, whereas all the surviving mice were euthanized at the end of the experiments. The whole brain and liver were collected from the euthanized mice and frozen for RVFV RT-qPCR or, for the brain, fixed in 3.7% formaldehyde for immunohistochemistry analyses. The blood sera (250 µL/sampling) were collected at D0, D3, D6, and D10 p.i. Each experiment was performed independently twice for each viral strain, and in two different laboratories (ANSES and Institut Pasteur in Paris). The data from these two experiments were similar and merged. Statistical analyses (Gehan–Breslow–Wilcoxon test) and Kaplan–Meier survival plots were conducted using Graphpad Prism 8.4.

### 2.7. RVFV RT-qPCR

The collected blood sera (100 µL) were directly mixed with 400 µL of AVL buffer from the QIAmp Viral RNA kit (Qiagen, Courtaboeuf, France). The whole brain and liver of euthanized mice were weighed and homogenized in a 500 µL DMEM using Tissue Lyser II (Qiagen). The viral RNAs were purified using the QIAmp Viral RNA kit (Qiagen) following the manufacturer’s protocol. The detection of the RVFV M segment was conducted by RT-qPCR using the SuperScript III Platinum One-Step qRT-PCR kit (Thermo Fisher Scientific, Villebon-sur-Yvette, France), as described elsewhere [35,36]. The efficiencies were estimated from the standard curves based on ten-fold dilutions of each viral stock, and were used to convert the Ct values to PFU-per-milliliter equivalents (eqPFU/mL) for the sera, or PFU-per-gram equivalents (eqPFU/g) for the tissues. Wilcoxon–Mann–Whitney test analyses were conducted using R software.

### 2.8. RVFV Serological Assays (ELISA)

The sera from the SB- and Cl.13-infected mice were tested for the presence of anti-RVFV IgG and IgM antibodies with in-house ELISA assays, as described previously [35,37]. Briefly, the IgG antigens were prepared from the MP-12 RVFV-infected cells while the IgM antigens were made from the MP-12 RVFV-infected cells and their supernatants. The negative controls without antigens correspond to the uninfected cells treated like the previous ones.

For the in-house IgG ELISA assay, 96-well culture plates Nunc MaxisorpTM (Thermo Fisher Scientific, Villebon-sur-Yvette, France) were coated with 100 µL of RVFV antigens (diluted 1:500 in PBS 0.01% Sodium Azide, Sigma-Aldrich, Merck, Saint-Quentin Fallavier, France) or negative-control antigens, and incubated overnight at 4 °C. The plates were then washed three times with 300 μL of PBS 0.05% Tween20 (PBS-T) (Euromedex, Souffelweyersheim, France), and the individual mouse sera were diluted 1:100 in PBS-T 5% skimmed milk powder (*w*/*v*) (PBS-T-M) and incubated at 37 °C for 1 h on coated plates. The plates were then washed three times with PBS-T and incubated at 37 °C for 1 h with HRP-conjugated rabbit anti-mouse IgG (whole molecule) (SigmaA9044) diluted 1:5000 in PBS-T-M. Finally, the plates were washed six times with PBS-T and the horseradish peroxidase (HRP) activity was revealed using the 3,3′,5,5′-tetramethylbenzidine (TMB) substrate (Life Technology). The chromogenic reaction was interrupted by adding 10.6% orthophosphoric acid (100 μL/well) (Alfa Aesar, Thermo Fisher Scientific, Villebon-sur-Yvette, France). The optical density (OD) was measured at 450 nm using a TECAN microplate reader. The adjusted OD450 values were calculated by subtracting the OD450 value of the negative Ag-coated wells (background) from that corresponding to those coated with the RVFV IgG antigens.

An in-house anti-RVFV IgM ELISA assay was developed using the capture ELISA method. Briefly, the micro-titer plates (Maxisorp, Nunc, Thermo Fisher Scientific, Villebon-sur-Yvette, France) were coated with a rabbit anti-mouse IgM (μ-chain specific) antibody (SAB3701197 Sigma-Aldrich, Merck, Saint-Quentin Fallavier, France) (100 μL/well, diluted 1:400 in PBS 0.01% Sodium Azide), and incubated overnight at 4 °C. The plates were then washed three times with PBS-T and the mice sera (100 μL/well, diluted 1:100 in PBS-T-M) were added into two adjacent wells (even- and odd-numbered columns). After 1 h at 37 °C, the plates were washed with PBS-T and coated with 100 μL/well of the RVFV antigens (even-numbered columns) or negative-control antigens (odd-numbered columns). After 1 h of incubation at 37 °C, the cells were washed three times and incubated for 1 h at 37 °C with 100 μL/well of hyperimmunized sera from a hamster infected with the ZH501 RVF strain (dilution 1:2000 in PBS-T-M). The cells were then washed three times and incubated for 1 h at 37 °C with 100 µL of a goat anti-hamster IgG (H + L)-HRP conjugated antibody (1:3000 in PBS-T-M) (Thermo Fisher Scientific). The OD was measured and calculated as described above for the RVFV IgG.

### 2.9. Immunohistochemistry (IHC)

The brain tissues of the infected or non-infected mice were embedded in paraffin and cut into 3 μm thick sections. The sections were treated with a BOND Epitope Retrieval Solution 1 (ready to use, citrate-based pH 6, epitope retrieval solution, AR9961, Leica Biosystems, Nanterre, France). An IHC detection of the RVFV-infected cells was performed with a polyclonal rabbit antibody raised against RVFV N-protein [38] diluted at 1:2000 in a Bond TM Primary Antibody Diluent (Leica Biosystems, Nanterre, France). The immunoreactive cells were revealed with a biotinylated goat anti-rabbit IgG secondary antibody at a 1:600 dilution (E0432, Dako, Agilent, Les Ulis, France). The biotin signal was detected using a Bond Intense R Detection System (DS9263, Leica Biosystems, Nanterre, France). The morphology of the microglial cells was assessed by immunohistochemistry using a rabbit anti-Iba1 primary antibody (#01919741, Wako chemical, Neuss, Germany, dilution 1:1000), as previously described [39]. The IHC analyses were conducted on consecutive sections and included the appropriate negative controls (an omission of the first antibody and non-infected brain tissues).

## 3. Results

### 3.1. The RVFV SB and Cl.13 Strains Are Lethal for Mice When Administered Intranasally

In order to assess whether the mortality rate depends on the route of administration, we inoculated BALB/c immunocompetent mice with 1 × 10^3^ PFU of either an RVFV SB or Cl.13 live-attenuated strain by subcutaneous (SC) or intranasal (IN) routes. We observed that 60% of the SB-IN-infected mice died between days 5 and 7 post-infection (p.i.), while 40% of them survived until day 15 p.i. (Figure 1A). The kinetics of the mortality of the Cl.13-IN-infected mice were slightly delayed compared to the SB and reached 65% on day 15 p.i. (Figure 1B). It is of note that the SC inoculation induced a very low mortality (only 1/17 mice with SB; *p* < 0.001 for both viruses, Figure 1). We concluded that an IN administration leads to a higher mortality rate compared to an SC inoculation, regardless of the attenuated RVFV strain used.

### 3.2. Intranasal Exposure of SB Induces Higher Viral Load Than the SC Inoculation

We then measured by the RT-qPCR the serum viral load of the mice who had been IN or SC infected with the SB or Cl.13 strain at D3, D6, and D10 p.i. The SB-inoculated mice displayed significantly higher viral loads after the IN than SC inoculation at D3 and D6 p.i. (Figure 2A; *p* < 0.0001 or *p* < 0.05). By contrast, the Cl.13-infected mice showed a very low and inconsistent serum viral load at all the time points (Figure 2B). 

Finally, we evaluated the immune response against an SB or Cl.13 infection in the IN- or SC-infected mice by measuring the titers of the anti-RVFV IgM and IgG by an ELISA at D0, D3, D6, D10, and D15 p.i. We found that, regardless of the strain and the route of administration, all the mice raised anti-RVFV IgM antibodies by D6, while the first IgG-positive mice were detected on day 10, with a higher frequency in SB- than in Cl.13-infected mice (12/17 vs. 1/12, respectively, Fisher’s test *p* = 0.0018; Table 1). These results indicate that both SB and Cl.13 induce an immune response against RVFV.

### 3.3. SB and Cl 13 Strains Are Detected in the Brain of IN-Infected Mice

To study the dissemination of the SB and Cl.13 infections, we measured the viral loads in the whole brain and liver of IN-infected mice that were moribund and euthanized between D5 and D7 p.i. for SB and D6 and D13 p.i. for Cl.13. Notably, while both strains were detected in the two organs, the viral loads were significantly higher in the brain compared to the liver (log_10_ (eqPFU/g) = 11 vs. 2 for SB and 7 vs. 3 for Cl.13, respectively, *p* < 0.0001 and *p* < 0.05; Figure 3). 

These results indicate that SB and Cl.13 were able to disseminate and replicate more efficiently in the brain than in the liver of IN-infected mice. Notably, this difference in the viral loads was much higher for SB- than Cl.13-IN-infected mice (9 vs. 4 log_10_ (eqPFU/g)). These findings were further confirmed by an immunohistochemistry assay using an antibody raised against the RVFV N-protein. A positive cytoplasmic labeling was detected in several areas of the brain of both the Cl.13- and SB-IN-infected mice (Figure 4) and we did not identify virus-specific labelled areas. Interestingly, positive cells were often clustered and less frequently scattered, with morphological features typical of neurons of different sizes and of glial cells (Figure 4a,c). Immunolabelling with an anti-Iba1 antibody revealed the marked activation of microglial cells in the Cl.13-infected mice (Figure 4d), while the activation was mild in the SB-infected mice (Figure 4b).

### 3.4. RVFV SB Attenuated Strain Is Highly Replicative in Human Neural and Liver Cells

To further characterize the SB and Cl.13 strains, we assessed their replication in human pluripotent stem cells differentiated into neural cells (neural hIPCs) or human hepatocytes (HepaRG) at 24 h and 48 h p.i. SB displayed much higher viral titers than Cl.13 in both of the cell types and at both time points (Figure 5), suggesting that the SB strain has a stronger replication capacity than Cl.13.

## 4. Discussion

In this study, we aimed to assess the virulence and replication kinetics of two RVFV live-attenuated vaccines—the SB and Cl.13 strains—using SC and IN routes of inoculation. We showed that both strains are virulent in immunocompetent mice when administered intranasally. The SB-infected mice succumbed more rapidly to infection than the Cl.13-infected mice, although the final survival rate was similar (40% vs. 35%) for both of the strains. The difference in the lethality kinetics could be explained by the higher replicative capacity of the SB compared to Cl.13. Indeed, we found higher viral loads in the sera and brain of the SB-IN- compared to Cl.13-IN-infected mice, and higher viral titers in the hIPSCs and HepaRG immunocompetent cells infected with the SB strain compared to those infected with the Cl.13 strain. The reduced in vitro and in vivo replicative capacity of Cl.13 and its rapid clearance from the sera of immunocompetent mice is probably due to the absence of a functional NSs protein, a key RVFV virulence factor that blocks the host innate immune pathway [20,40]. 

Interestingly, both strains displayed higher viral titers in the brain than in the liver of IN-infected mice. This could be explained by the ability of RVFV to target the neurons lining the nasal tract that, in turn, could give the virus direct access to the brain [26,27,28,29]. From there, both strains may then reach the liver where either their attenuated status and/or the rapid mice death could impair their efficient replication in this organ, thereby explaining their reduced viral titers. On the other hand, one could raise the hypothesis that the reduced replication activities of SB and Cl.13 are also due to the stronger innate immune control of the viral infections exerted by the liver compared to the brain [41]. Indeed, although the SB encodes a functional NSs (unlike Cl.13), its attenuation status, yet uncharacterized, may affect its replication capacity in an immunity-independent manner. This, combined with the strong innate immunity of the liver, could prevent an efficient viral replication of SB in this organ and not in the brain.

Moreover, since the SB strain was generated by multiple passages in the mouse brain, it is likely to be more neuro-adapted than Cl.13, which could explain why the viral loads between the brain and the liver are much higher in the SB- than Cl.13-IN-infected mice (9 vs. 4 log_10_ (eqPFU/g)). Previous studies have shown that, in several animal models, an aerosol or IN exposure to RVFV leads mainly to severe neurological symptoms compared to peripheral infection routes [27,29,31,42,43,44,45,46]. However, most studies have uses virulent RVFV strains at infectious doses that induce rapidly developing severe disease and death, making it difficult to study the impact of administration routes on the virulence and pathogenicity. The use of live-attenuated strains derived from pathogenic RVFV strains (e.g., SB, Cl.13, and MP12) with a reduced virulence and pathogenicity provide better experimental models for such comparisons [47,48,49]. 

Many of the current RVFV live-attenuated vaccines induce adverse effects. For this reason, several laboratories have started developing safer candidate vaccines using reverse genetic systems. For example, Wichgers Schreur et al. developed a candidate vaccine based on a four-segmented Rift Valley fever virus (vRVFV-4s) that does not induce mortality in mice infected by an IN exposure, unlike SB or Cl.13 [32,50]. Dodd and colleagues demonstrated that a single NSs deletion RVFV mutant (ΔNSs rRVFV) causes lethal encephalitis by an IN exposure [31], whereas Bird et al. showed that a double deletion RVFV lacking the NSs and NSm virulent genes (ΔNSs-ΔNSm rRVFV) appears safe and effective when inoculated subcutaneously, even though its virulence has not been tested by an IN exposure yet [49]. Overall, these and our data highlight the importance of investigating the pathogenicity and, more particularly, the neurovirulence of RVFV live-attenuated vaccines using a different route of transmission to ascertain their safety.

Finally, our study revealed that both SB and Cl.13 can infect the neurons and microglia in the brain of infected mice, in accordance with what was previously observed with other RVFV strains [22,51,52]. Microglia cells are one of the main type I IFN-producing cells, an important first line of defense against viral infections [53] and the lack of the NSs protein in Cl.13 reduces its ability to antagonize the host immune response [20,40]. Interestingly, we observed a pronounced IBA-1 labeling in the brain of the Cl.13-IN-infected mice, suggesting a significant activation of the microglia cells in response to the infection. Therefore, the active replication of Cl.13 in the brain together with its lack of a functional NSs protein may lead to a strong neuroinflammation that, in turn, could participate to virus-induced neuropathogenesis in Cl.13-IN-infected mice [54]. Overall, SB displays a high replication capacity and the ability to antagonize the IFN response and, thus, the microglial activation. This, in turn, may lead to a rapid neurological disease and the death of the infected animals. On the other hand, Cl.13 may replicate less efficiently than SB but cause a protracted central nervous system disease which is associated with neuroinflammation. Further brain anatomopathological and neuroinflammation analyses over the course of the SB and Cl.13 infection will certainly answer these questions.

Our study shows that both the SB and Cl.13 attenuated vaccine strains are lethal only when administered intranasally (and not subcutaneously) in immunocompetent mice. These findings carry important scientific implications not only for a better understanding of the correlation between viral pathogenicity and the route of administration of RVFV, but also for raising awareness during vaccination campaigns against RVFV in Africa, where both strains are routinely used during epizootics [17]. It is indeed of the outermost importance to take special precaution to contacts and aerosols during these practices.

## Figures and Tables

**Figure 1 viruses-14-02470-f001:**
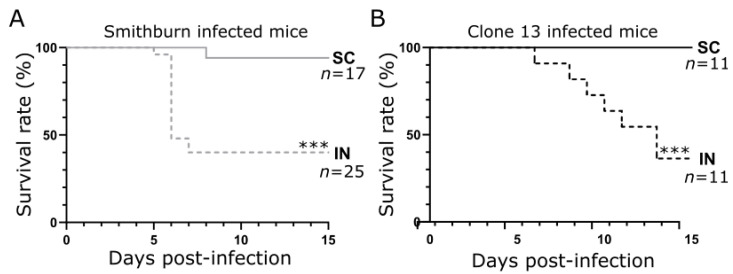
Survival rates of BALB/c mice intranasally or subcutaneously infected with the SB or Cl.13 vaccine strain. Kaplan–Meier survival plots of BALB/c mice subcutaneously (SC) or intranasally (IN) inoculated with 1 × 10^3^ PFU of the Smithburn (**A**) or Clone 13 (**B**) strain. For both RVFV strains, the IN route of infection led to higher mortality rates than SC inoculation. Survival curves were compared using the Gehan–Breslow–Wilcoxon test (***: *p* < 0.001). n: number of mice per condition from, at least, two independent experiments.

**Figure 2 viruses-14-02470-f002:**
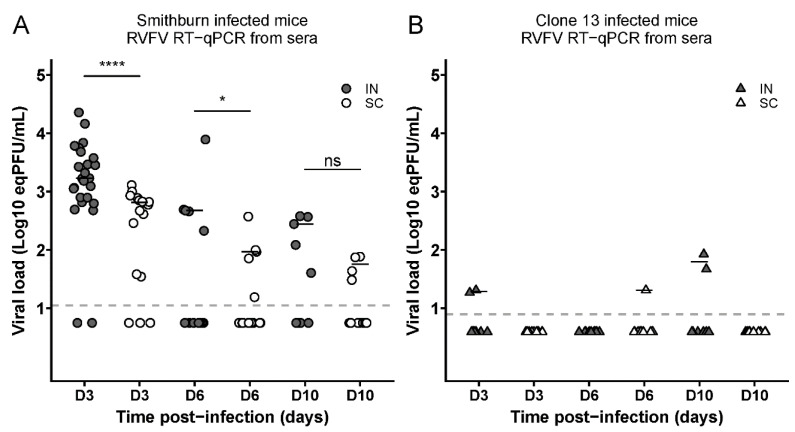
Intranasally SB-infected mice display higher viremia than those subcutaneously infected. The graphs report the viral loads (expressed as eqPFU/mL) in the sera of intranasally (IN)- or subcutaneously (SC)-infected mice with the SB (**A**) or Cl.13 (**B**) strain, collected at D3 (n: SB-IN = 25; SB-SC = 17; Cl.13-IN = 11; and Cl.13-SC = 11), D6 (*n*: SN-IN = 15; SB-SC = 17; Cl.13-IN = 11 and Cl.13-SC = 11) and D10 (*n*: SN-IN = 10; SB-SC = 16; Cl.13-IN = 11; and Cl.13-SC = 11) p.i. Dashed lines indicate the threshold of virus detection [log_10_ (eqPFU/mL) = 1.049 for SB and 0.895 for Cl.13]. Negative sera are shown below these lines for each virus at half the value of their detection limits. For SB data, statistical analyses were performed using Wilcoxon–Mann–Whitney test (*: *p* < 0.05; ****: *p* < 0.0001 and ns: not significant).

**Figure 3 viruses-14-02470-f003:**
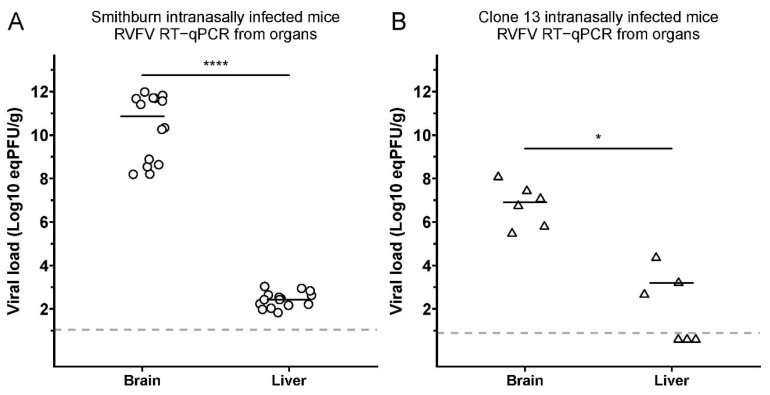
SB and Cl.13 display a higher viral load in the brain than in the liver of IN-infected mice. The graphs report the viral loads (expressed as eqPFU/g) in the brains and livers of euthanized mice intranasally (IN) infected with SB ((**A**); *n* = 15) or Cl.13 ((**B**); *n* = 6). Mice were euthanized when moribund between D5 and D7 p.i. for SB and D6 and D13 p.i. for Cl.13. Negative sera are shown below these lines for each virus at half the value of their detection limits. Statistical analyses were performed using Wilcoxon-Mann–Whitney test (*: *p* < 0.05; ****: *p* < 0.0001).

**Figure 4 viruses-14-02470-f004:**
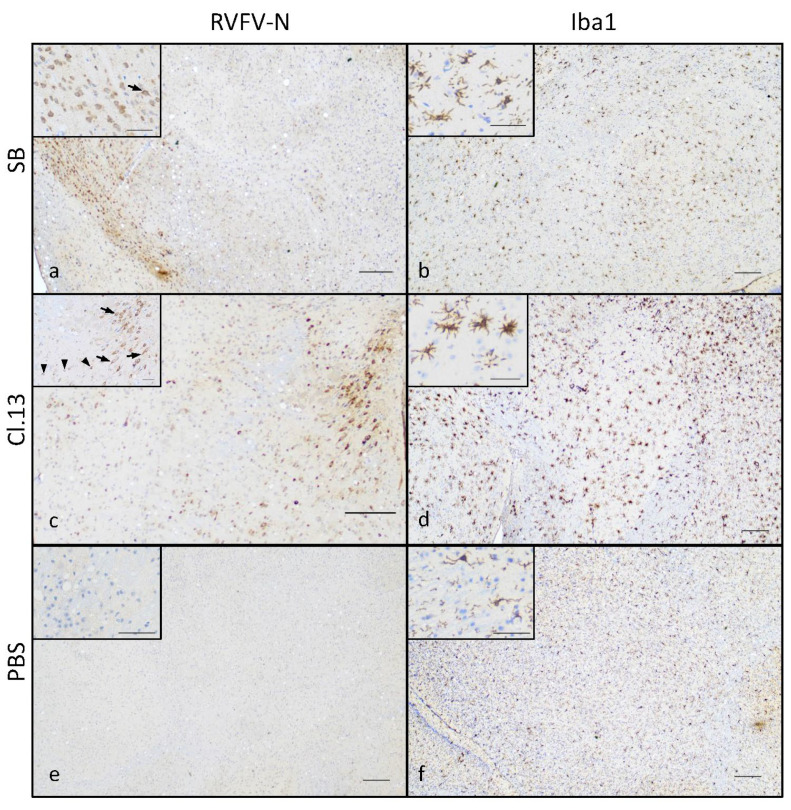
Brain of SB- and Cl.13-IN-infected mice displays N-protein-positive cells and activated microglia. Mice intranasally infected with the SB (**a**,**b**) or Cl.13 (**c**,**d**) RVFV strains, or inoculated with PBS (**e**,**f**) were euthanized at day 6–10 (Cl.13) or day 5–7 (SB) p.i. when moribund, or day 20 post-inoculation (PBS) and their brain was analyzed by immunohistochemical staining against RVFV N (**a**,**c**,**e**) and Iba1 (**b**,**d**,**f**) proteins. Representative low magnification (×4) images show immunoreactivity to RVFV N-protein in the cranioventral portion of the brain of SB-infected mice (**a**) and the midbrain of Cl.13-infected mice (**c**) and compared to uninfected controls (**e**). Higher magnification insets (×20) show cells with morphological features of neurons (arrows) and glial cells (arrowheads). Microglial reactivity assessed by Iba1 staining was mild in SB-infected mice (**b**) and more pronounced in Cl.13-infected mice with activated microglial cells ((**d**), inset). Bars: 200 µ in low magnification images and 50 µ in high magnification insets.

**Figure 5 viruses-14-02470-f005:**
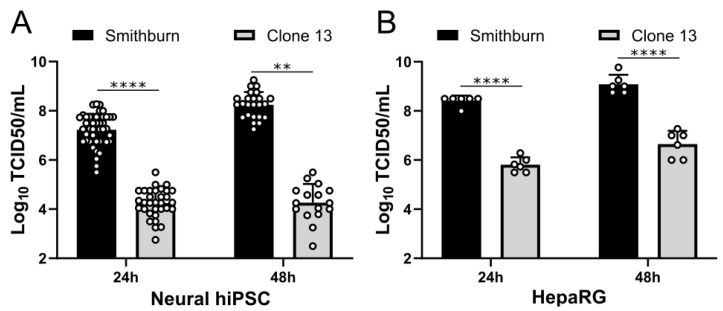
SB displays a high replication capacity in human neural and hepatocyte cells. Growth curves experiments were performed in neural-differentiated human pluripotent stem cells (neural hIPSC; MOI = 0.1) (**A**) and human hepatocytes cell line (HepaRG; MOI = 0.001) (**B**) infected with SB and Cl.13. Cell supernatants were collected at 24 h and 48 h p.i., and virus titers were obtained by limiting dilution assays on BSR cells and expressed as log_10_ 50% tissue culture infective doses (TCID50)/milliliter. Circles represent the viral titers obtained from independent experiments performed in triplicate and repeated three times. Bars indicate standard deviations. Statistical analyses were performed using Kruskal–Wallis (**: *p* < 0.01; ****: *p* < 0.0001).

**Table 1 viruses-14-02470-t001:** Seroconversion of intranasally and subcutaneously infected mice. The presence of anti-RVFV IgM and IgG antibodies in SB- and Cl.13-infected mice was detected by ELISA assays at D0, D3, D6, D10, and D15 p.i. The table indicates the number of IgM or IgG positive mice as well as the total number of mice analyzed at each time point and condition.

		D0		D3		D6		D10		D15	
	Ig Anti-RVFV	IgM	IgG	IgM	IgG	IgM	IgG	IgM	IgG	IgM	IgG
Exp. Conditions											
SB-IN		0/12	0/12	0/12	0/12	7/7	0/7	5/5	5/5	5/5	5/5
SB-SC		0/12	0/12	0/12	0/12	12/12	0/12	12/12	7/12	11/11	10/11
Cl13-IN		0/6	0/6	0/6	0/6	6/6	0/6	6/6	1/6	5/5	3/5
Cl13-SC		0/6	0/6	0/6	0/6	6/6	0/6	6/6	0/6	6/6	4/6

## Data Availability

The data that support the findings of this study are available upon request.

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
