# Peer review of "Intranasal Exposure to Rift Valley Fever Virus Live-Attenuated Strains Leads to High Mortality Rate in Immunocompetent Mice"

_viruses, 2022, doi:10.3390/v14112470_

Round 1

Reviewer 1 Report

The manuscript by Lacote and colleagues investigates the important role that inoculation route plays in neurovirulence in mice by two attenuated RVFV vaccine strains that are currently used in Africa for control and prevention of RVF outbreaks. This work is of high interest considering WHO’s designation of RVF as a high priority disease and the many ongoing efforts to develop safer attenuated live virus vaccines to prevent RVF outbreaks. The study findings highlight the importance of the route of exposure to RVFV vaccines. Moreover, the results suggest that intranasal inoculation of mice with candidate RVF vaccines may provide a useful assessment of vaccine attenuation and safety (especially potential for neurovirulence).

Comments

There are many typographical and grammatical errors throughout the manuscript that should be addressed.

Methods section 2.4. Details are lacking for the assays used to titrate. Brief descriptions should be included with appropriate citations referring to any methods described in greater detail by others.

Line 169 - TCID50 titration would seem to fit better in 2.4. Also, TCID50 not defined.

Methods section 2.6 lacks detail. How were mice inoculated? How were the mice anesthetized and in how much volume was the instillation of virus into the nares? For SC, was it footpad? Somewhere else? Were those mice also anesthetized? Could anesthetics affect vascular permeability and dissemination of the virus?

Line 180 - 12% doesn't seem like excessive/severe weight loss. Unclear if weight loss is relative to peak weight of the mouse, or the weight of the animal at the time of infection?

Lines 191-192 - More details are needed. Primer/probe sequences? Reaction conditions? Specifics on conversion of Ct to PFU equivalent (why not genome equivalents?)?

Methods section 2.9. I'm not sure this description is detailed enough. Please consider the important steps and add detail. Also, indicate reagents/kit used to detect biotin labeling of immunoreactive cells.

Fig. 1. Why end the study at day 15 when mice were dying out to day 13? Were the survivors in good health and gaining weight? Is it possible that other mice may have succumbed if the study went 21 days? Could mean group or individual animal curves be shown as supplementary data to reflect that the surviving mice were gaining weight or stable and would not be expected to die?

Fig. 2. No A or B labeling on the graph.

Fig. 2. Legend: Lines 268-269 - what is SN-IN?

Lines 275-280 - Actual Ab titer or concentration would be more informative and should be presented. Longitudinal data graphs with concentration/titer plotted as individual values for each animal would make for nice visualization of the experiment results.

Results section 3.3 - Here and below for the brain IHC figure, where the mice euthanized when moribund or when showing initial clinical signs of disease? This should be made clear here or in the methods section.

Fig. 3. - The number of animals (data points) do not add up to the n indicated in the figure legend. Why only 3 data points for Cl.13 liver?

Fig. 4. - Why show different parts of the brain for this comparison showing N staining for the 2 viruses? Better to show representative staining of the same region for both viruses. It’s unclear if there were also differences in the region stained for Iba1? Could certain areas have a higher density of glial cells with heightened sensitivity to activation by infection?

Fig. 5. - TCDI50? Also, says each experiment was conducted in triplicate but unclear what all the small circles in the graphs represent?

Discussion - I realize the SB and Cl.13 vaccines are currently used, but a reference someone in the discussion to the safer vaccines that are in development (i.e., 4 segment virus, double deletion virus, other?) would be relevant here to inform the reader about progress towards safer vaccine options that could be evaluated by IN route and compare to the current SB and Cl.13 viruses by this useful measure for potential neurovirulence.

Line 339 - I would not characterize these as highly virulent. Perhaps moderately virulent or some other description. Highly virulent would equate to 100% lethal.

Lines 365-366 - Consider adding the CDC double deletion virus here. Has that virus been tested by IN or aerosol administration?

Lines 372-374 - Along these lines, is greater SB replication in the brain due to its ability to disrupt IFN response and limit activation of microglia? Thus, higher replication leads to more rapid CNS disease and death, whereas Cl.13 causes a more protracted disease associated with neuroinflammation?

Reviewer 2 Report

The manuscript is well-written and interesting. 

Abstract

Line 27 - I believe you mean high titer in the brain rather than viremia.

Introduction

Line 56 - Prolificacy is an unusual, although accurate word. Perhaps replace it with a more familiar word.

Methods

Line 144 - Hedge not Hedged.

Line 181 - There are no IHC results for the liver. This line makes it sound like both brains and livers were fixed for IHC.

Line 183 - How much blood/serum was collected at each time point? Was serum collected at D0? ELISA results show that it was.

Line 188 - Homogenizing serum seems odd. Was the serum also diluted in 500 ul of DMEM? If so how much serum was added? Or was the serum used directly in the RNA isolation?

Line 99 - Are the antigens used for the IgG and IgM ELISA's different? Wouldn't they both be RVFV proteins? Did you use cell lysates or were the antigens purified in some fashion?

Results

Figure 2 - The color of the IN inoculated mice is difficult to see. Perhaps the outline of the points could be made smaller or the color could be darkened. Please remove the background axis lines. They are just visible enough to be distracting.

Line 281 - Can you say the immune response was effective since more than half of the mice died and no testing of the antibodies was done for neutralizing ability?

Figure 3 - Please remove background axis lines. X-axis title is incorrect. If you are going to use different colors for brain and liver data points, a legend should be included. But because the labels on the X-axis describe the organ the data is from you don't really need different colors. Should the Y-axis title be eqPFU/g not per ml?

Figure 4 - Could the contrast be adjusted on the image? It is difficult to see detail. For consistency, the PBS should have an inset as well. But it is obvious there is no labelling. Please add the magnification of the images and insets to the caption. On line 318 cells are described as glial cells. Everywhere else they are called microglial cells. Why were different areas of the brain used for SB and Cl.13? Was this because infected cells were found in different areas for each virus? If so, is that important?  If infected cells were found in the same areas of the brain for both viruses why were images from different areas used?

Figure 5 - Please make the Y-axis the same for A and B and start it at 0. What is the immune status of the cells used? Are they immunocompetent? This could explain some of the variation between SB and Cl.13 at each time point.

Discussion

Paragraph 2 - Please explain the idea that the innate immunity of the liver is stronger than in the brain in more depth. SB, although attenuated, may still retain some ability to antagonize the interferon response similar to MP-12. The difference in activation of microglial cells between SB and Cl.13 would suggest that this is the case and is discussed in paragraph 3.

Line 359 - The viral loads in the liver according to Figure 3 for both viruses are very similar, not 5 log10 different.

Line 364 - It seems that your study resulted in rapid death also. Would you suggest that titration of the inoculation dose would allow better differentiation of virulence resulting from route of inoculation?

Have you considered what it means that Cl.13 is barely detected in serum at any time point yet it grows to a slightly higher titer in the liver than SB, which is seen in the serum at all times?

A few sentences on what is needed in future studies to identify pathology resulting specifically from route of infection would bolster the discussion.
